# Minimal Clinically Important Difference (MCID), Substantial Clinical Benefit (SCB), and Patient Acceptable Symptom State (PASS) of the Shoulder Disability Questionnaire (SDQ) in Patients Undergoing Rotator Cuff Repair

**DOI:** 10.3390/ijerph20115950

**Published:** 2023-05-25

**Authors:** Umile Giuseppe Longo, Rocco Papalia, Sergio De Salvatore, Carlo Casciaro, Ilaria Piergentili, Benedetta Bandini, Alberto Lalli, Edoardo Franceschetti, Vincenzo Denaro

**Affiliations:** 1Research Unit of Orthopaedic and Trauma Surgery, Fondazione Policlinico Universitario Campus Bio-Medico, Via Alvaro del Portillo, 200, 00128 Roma, Italy; 2Research Unit of Orthopaedic and Trauma Surgery, Department of Medicine and Surgery, Università Campus Bio-Medico di Roma, Via Alvaro del Portillo, 21, 00128 Roma, Italy; 3Laboratory of Measurement and Biomedical Instrumentation, Campus Bio-Medico University, Via Alvaro del Portillo, 200, Trigoria, 00128 Rome, Italy

**Keywords:** MICD, SCB, PASS, SDQ, rotator cuff repair

## Abstract

The Shoulder Disability Questionnaire (SDQ) is a Patient-Reported Outcome Measure (PROM) applied to evaluate shoulder surgery outcomes. The purpose of this study is to identify the accurate Minimal Clinically Important Difference (MCID), Substantial Clinical Benefit (SCB) and Patient Acceptable Symptom State (PASS) values for the SDQ score. A total of 35 patients (21 women and 16 men, mean age 76.6 ± 3.2 years) were followed up at 6 months postoperatively. To assess the patient’s health satisfaction and symptoms, anchor questions were used. The MCID and SCB values of the SDQ score for patients who underwent arthroscopic rotator cuff repair from inception to final follow-up were 40.8 and 55.6, respectively. A change of 40.8 in the SDQ score at 6 months after surgery shows that patients achieved a minimum clinically important improvement in their state of health, and a 55.6 change in the SDQ score reflects a substantial clinically important improvement. The PASS cut-off of the SDQ score at 6 months postoperatively ranged from 22.5 to 25.8. If an SDQ score of 22.5 or more is attained after surgery, the health condition can be recognized as acceptable by the majority of patients. These cut-offs will help with understanding specific patient results and allow clinicians to personally assess patient improvement after rotator cuff repair.

## 1. Introduction

Patient-Reported Outcomes (PROs) are the description of the patient’s health as reported directly from the patient [1], avoiding the objective interpretation of physicians [2]. In order to ensure a reliable measure of these results, we can apply specific tools [3]. The Patient-Reported Outcome Measures (PROMs) are questionnaires filled out directly by the patients, consisting of a given set of questions regarding the well-being of the patient [4]. The PROMs offer an in-depth knowledge of the patient’s whole experience, helping to decide the most suitable therapy and service for the patient [5].

Among the most frequent musculoskeletal complaints is shoulder pain [6]. Most cases of shoulder pain are caused by rotator cuff disease [7]. Only 21–50% of patients record an improvement in symptoms within six months after treatment [8]. For this reason, several PROMs are used to assess shoulder surgery outcomes: simplistically, these outcomes are exploited in the literature to determine the number of patients, enrolled within a single study, that achieve a certain threshold in their treatment outcomes, in order to determine the efficacy of a particular treatment. In particular, the Shoulder Disability Questionnaire (SDQ) is a common tool in clinical studies [9]. This PROM is typically used in orthopedic studies [10]. The SDQ is a questionnaire able to point out the restrictions of daily activities caused by the functional incapacity of the shoulder [11]. The SDQ investigates both clinically important parameters and the emotional well-being of the patient [12,13].

For the majority of patients, shoulder surgery results in considerable SDQ improvements with statistically significant alterations between the preoperative and postoperative follow-up [14]. The importance of assessments focused on the patient, which may be applied to understand surgical outcomes based on their therapeutic value to patients, has gradually increased in recent years [15,16].

A significant measure of variation in the outcome under research must be used in order to comprehend the differences between cohorts in clinical studies and determine the true impact of a specific therapy [17]. The Minimal Clinically Important Difference (MCID), the Substantial Clinical Benefit (SCB) and the Patient Acceptable Symptom State (PASS) can be applied to detect this change [18].

The MCID represents the lowest variation in the result that is able to provide clinical improvement after surgery [19,20]. The MCID is used in clinical studies as a tool to measure the magnitude of progress in a PROM that reflects medical significance [19].

The SCB is an alternative psychometric value that recognizes a degree of outcome amelioration that is identified as sufficient by the patient [21]. Both the MCID and the SCB can define meaningful improvement: the MCID detects the lowest important change and the SCB recognizes when that change has occurred [22].

The PASS signifies the magnitude of the postoperative result required for patient fulfillment [20]. The PASS differs from the MCID as it does not compare the preoperative and postoperative values of the PROM, but rather assesses a cross-sectional PROM value postoperatively [23].

The purpose of this study is to identify accurate MCID, SCB, and PASS values in patients with shoulder disease for the SDQ score.

## 2. Materials and Methods

This study investigated data retrieved from patients who underwent arthroscopic rotator cuff repair for rotator cuff tears of any grade between September 2019 and March 2020 in Campus Bio-Medico of Rome. Overall, 35 patients (21 women and 16 men, mean age 76.6 ± 3.2 years) were enrolled and followed up at 6 months after surgery. No patients were lost to follow-up. The SDQ score was assessed preoperatively and at 6 months postoperatively. Patient’s health satisfaction and symptoms at the final follow-up were evaluated by anchor questions.

### 2.1. The Shoulder Disability Questionnaire (SDQ)

The SDQ is a pain-related disability questionnaire, which comprises 16 questions referring to the previous 24 h daily activities that may cause pain in patients with shoulder illnesses. Possible answers are “Yes” (=0), “No” (=1), or “Not applicable” (=missing value). The “Not applicable” response should be given when the patient did not perform the activity with the last 24 h. The final score for each patient is calculated by multiplying the score by 100 and dividing the total number of “Yes” responses by the total number of applicable items. The total score ranges from a minimum of 0 (best condition: no disability) to a maximum of 100 (worst condition) [14,24]. The internal consistency and test–retest reliability were supported (α = 0.78 and ICC = 0.97 [95%CI = 0.96–0.98]). More than 75% of the anticipated correlations were reached for each subscale, demonstrating construct validity [12]. Moreover, the SDQ did not show ceiling and floor effects [12]. The average time needed to answer all questions was 2.30 min (range 1.50–3.10 min). No multiple replies or missing responses were found [12].

### 2.2. Anchor Questions

At the final evaluation, the patients completed the anchor questions. In order to identify an outcome change that was clinically significant, these questions served as anchors. “Would you generally think that your health is at least good?” was the anchor question applied to determine the PASS scores. The potential responses were “Yes” or “No”. “How do you feel following the surgical procedure?” was asked to calculate the MCID and SCB cut-offs. The 5-point global scale used for scoring responses varied from −2 (“far worse”) to +2 (“much better”). Patients who answered “much worse”, “a little worse”, or “equal” had a value of −2 to 0 and corresponded to the “No change group”. The patients who answered “a little better” represented the “Minimal improvement group”. Patients who answered “much better” represented the “Substantial improvement group”. The MCID cut-offs were computed comparing the “No change group” with the “Minimal improvement group”, while comparing the “No change group” with the “Substantial improvement group” and the SCB values were calculated. The satisfied patients were compared with the unsatisfied in order to calculate the PASS thresholds. The MCIDs were also computed using the distribution techniques.

### 2.3. Statistical Analysis

To perform the statistical analysis, IBM SPSS Statistics for Windows, Version 26.0. (Armonk, NY, USA: IBM Corp) was utilized. An priori power analysis was conducted using G*Power. With an alpha level of 0.05, minimum power established at 0.80, and an effect size (derived by dividing the mean of the change in the patients by the standard deviation of the differences among the patients) of 0.94 (95% CI: 0.52–0.98) [25], given the lowest bound of the confidence interval, finding a statistically significant impact needs a minimum of 25 patients. Data normality for the SDQ score using the Shapiro–Wilk test was tested. The normality of the SDQ score, at inception and final follow-up at 6 months postoperative, were compared using the paired t-test. The level of statistical significance was 0.05. The distribution methods used to find the thresholds of the MCID were 0.5 SD (one-half of the standard deviation), the SEM (standard error measure) and the MDC (minimal detectable change). The 0.5 SD was calculated as the half of the standard deviation. According to the research conducted by Norman et al., the MCID in many experiments had a value of 0.5 SD. They added that for a reliability of 0.75, 0.5 SD is equal to 1 SEM [26]. The formula used to calculate the SEM was SD*√(1 − α), where α represents Cronbach’s alpha, the value of the consistency of the SDQ. The MDC formula was 1.96*SD*√2. The 0.5 SD was connected to the median effect size. The SEM constitutes the lowest variation beyond the measurement error, and it represents the variation in scores due to the unreliability of the scale or measure used. It is more probable that a change smaller than the determined SEM is due to measurement error than it is due to a real change that was witnessed. When the MCID was determined using the conventional anchor-based approach, Wyrwich et al. discovered that a value of 1 SEM matched the MCID value [27]. The MDC constitutes the lowest variation beyond the measurement error with a 95% confidence interval, so a valid MCID should then be at least as large as the observed MDC [28]. To evaluate the cut-off values of MCID and SCB change, based on the anchor method calculation, the receiver operating characteristic (ROC) curve and mean change (MC) were applied. The ROC curve and the 75th percentile of the cumulative percentage curve of patients who believe their symptoms are under control (patients who answered “Yes” on the questionnaire) were used to determine the PASS threshold [28,29]. The ROC curve is the graph of sensitivity versus (1 − specificity) as the cut-point runs through the whole range of possible values. The threshold was determined with the Youden index (sensitivity + specificity − 1), which optimizes the percentage of correctly classified items [30]. The acceptable values of the area under the roc curve (AUC) were ≥0.7 [31]. The MC method found the thresholds of the MCID (or the SCB) through the mean difference from initial to final evaluation score in the “Minimal improvement group” (or “Substantial improvement group”) [32]. The ceiling and floor effects were calculated. Ceiling and floor effects happen when a significant proportion of subjects receive the highest or lowest scores, respectively, making it impossible for the measure to distinguish between subjects who fall at either ends of the spectrum. A ceiling or floor impact is usually described in the orthopedics community as 15% (or more) of participants in a group scoring at the highest or lowest level [33,34,35].

## 3. Results

The normal distribution of the SDQ score with the Shapiro–Wilk test was assessed (*p* = 0.4). The baseline SDQ score was 69.3 ± 25.2 (range: 7.7–100) and the postoperative 6-month SDQ score was 15.7 ± 18.3 (range: 0–61.5). At the preoperative follow-up, the floor effect was 0% and the ceiling effect was 9.1%. At the postoperative follow-up, the floor effect was 29.5% and the ceiling effect was 0%. A statistically meaningful mean change from initial to final evaluation was achieved (*p* < 0.001).

Medium–high internal consistency for the SDQ score was found (α = 0.7). The MCID values computed by distribution methods were 13.1 with the 0.5 SD method, 14.7 with the SEM approach and 40.8 with the MDC technique. The MCID cut-offs assessed using the anchor methods were 23.6 (AUC = 0.6) with the ROC method and 48.5 with MC approach. The SCB thresholds found were 40.4 (AUC = 0.6) with the ROC method and 55.6 with MC approach. The PASS cut-off calculated with the ROC method was 25.8 (AUC = 0.7) and 22.5 with 75th percentile approach (Table 1).

## 4. Discussion

The principal finding of this research is that, following arthroscopic rotator cuff surgery, the MCID and SCB values of the SDQ score were 40.8 and 55.6, respectively, from the initial to the final assessment. Many clinical studies have been carried out for the MCID, SCB or PASS values in patients with shoulder conditions. The MCID, SCB and PASS thresholds for the Single Assessment Numeric Evaluation (SANE), the American Shoulder and Elbow Surgeons (ASES) and Constant scores at the 1-year interval in patients who underwent shoulder arthroplasty were found by Gowd et al. [36], with the SANE score being the only score to have acceptable an AUC for achieving an MCID with a cut-off of 28.8. Cvetanovich et al. [37] reported the MCID, SCB and PASS cut-offs for patients undergoing arthroscopic rotator cuff repair that were, respectively, 11.1, 17.5, and 86.7 for the American Shoulder and Elbow Surgeons (ASES) score. Berthold et al. [38] calculated the MCID, SCB and PASS cut-offs for the ASES, Simple Shoulder Test (SST) and Visual Analogue Scale (VAS) pain scores after total shoulder arthroplasty comparing the dominant versus the non-dominant side, without a statistically significant difference in the MCID (85% vs. 93%, respectively), SCB (54% vs. 73%, respectively), or PASS (50% vs. 71%, respectively) thresholds.

No author has ever calculated the MCID, the PASS and the SCB for the SDQ. To the knowledge of the authors, this is the first research to accomplish a calculation of the MCID, the SCB and the PASS values for SDQ. For this reason, this article’s purpose is to find the MCID, SCB and PASS values for the SDQ score after 6 months of rotator cuff repair.

In this article, different methods to calculate the MCID’s cut-offs were used—three distribution methods (0.5 SD, SEM and MDC) and two anchor-based methods (ROC and MC). There is no agreement in the literature about which technique is the most appropriate. The MCID thresholds found in this study ranged between 13.1 and 48.5. According to Copay et al. [28], a change lower than the detected SEM is likely the result of miscalculation rather than a genuine variation. For this reason, the MCID calculated with the 0.5 SD method is not valid because it is less than the SEM. Furthermore, since the AUC was a less than acceptable threshold of 0.7 [28] (AUC = 0.6), the ROC and the MC anchor approaches are not appropriate. The ability to correctly discriminate all the patients is not satisfied. Given the above, the MDC method is the most appropriate MCID value. Therefore, the MCID value for the postoperative SDQ score after 6 months is 40.8.

To calculate the SCB thresholds, both the ROC and MC anchor approaches were used. The AUC was less than 0.7 (AUC = 0.6). The SCB value of the postoperative SDQ score after 6 months ranged between 40.4 with ROC and 55.6 with the MC method. Since the SCB cut-off calculated with the ROC method is less than the MCID, the SCB value of the postoperative SDQ score after 6 months is 55.6.

The PASS cut-offs with both the ROC and 75th percentile methods were calculated. The AUC was acceptable (AUC = 0.7). The PASS values ranged from 22.5 with the 75th percentile approach to 25.8 with the ROC technique.

At the preoperative follow-up, the floor effect was 0% and the ceiling effect was 9.1%. At the postoperative follow-up, the floor effect was 29.5% and the ceiling effect was 0%. No ceiling effect was found. The floor effect was found at 6 months follow-up.

This study has various strengths. To the knowledge of the authors, this is the first article in the current literature that calculates the MCID, SCB and PASS thresholds of the SDQ score for patients who received arthroscopic rotator cuff repair, reflecting a need to find precise values that can be applied in clinical practice. Furthermore, using only validated methods to evaluate the MCID, SCB and PASS scores increases the reliability of this study’s outcomes. Additionally, to obtain reliable findings, the MCID was calculated applying both the distribution and anchor approaches.

However, this study presents some limitations. Since the authors decided to apply only one anchor, the uniformity of outcomes across other anchors was not computed. Moreover, at the final 6-month evaluation, the MCID, SCB, and PASS cut-offs were assessed, resulting in a lack of information regarding the long-term outcomes. Finally, even if the number of patients enrolled is enough according to the a priori power analysis, a greater study cohort is typically used in the current literature regarding this topic. In order for the current findings to be generalizable and applied by the researchers, a larger cohort is required. Additionally, the generalizability of the MCID, the PASS and the SCB has recently been addressed and questioned, since, among others, Wellington et al. showed that there is no homogeneity between the SCBs of four different commonly exploited shoulder outcomes when separating patients by time or geographic region regarding shoulder arthroplasty [39].

Finally, an easier and more commonly validated scale such as the Likert scale may have been exploited to grade the patients’ answers to the anchor questions. The exploited scale was used to avoid inter-rater variability; however, a certain degree of confounding could not be avoided given the vulnerability of the exploited scale and the dependence on the clinician’s assessment [40].

## 5. Conclusions

After arthroscopic rotator cuff repair, the MCID and SCB values of the SDQ score from initial to final evaluation at 6 months were 40.8 and 55.6, respectively. A change of 40.8 in the SDQ score at 6 months after surgery shows that patients achieved a minimum clinically important improvement in their state of health, and a 55.6 change in the SDQ score reflects a substantial clinically important improvement. The PASS cut-off of the SDQ score at 6 months postoperatively ranged from 22.5 to 25.8. If an SDQ score of 22.5 or more is attained after surgery, the health condition can be recognized as acceptable by the majority of patients.

These cut-offs will help with understanding specific patient results and allow clinicians to personally assess patient improvement after rotator cuff repair.

## Figures and Tables

**Table 1 ijerph-20-05950-t001:** MCID, SCB and PASS outcomes.

	Value	Method	Anchor
MCID	13.1	0.5 SD	No anchor
	14.7	SEM	No anchor
	40.8	MDC	No anchor
	23.6 (0.6)	ROC (AUC)	“How do you feel after the surgical intervention carried out?”
	48.5	MC	“How do you feel after the surgical intervention carried out?”
SCB	40.4 (0.6)	ROC (AUC)	“How do you feel after the surgical intervention carried out?”
	55.6	MC	“How do you feel after the surgical intervention carried out?”
PASS	25.8 (0.7)	ROC (AUC)	“In general, would you say that your health is at least good?”
	22.5	75th Percentile	“In general, would you say that your health is at least good?”

## Data Availability

The data presented in this study are available on request from the corresponding author.

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
