# Peer review of "Minimal Clinically Important Difference (MCID), Substantial Clinical Benefit (SCB), and Patient Acceptable Symptom State (PASS) of the Shoulder Disability Questionnaire (SDQ) in Patients Undergoing Rotator Cuff Repair"

_ijerph, 2023, doi:10.3390/ijerph20115950_

Round 1

Reviewer 1 Report

­          The authors present a well done and interesting article which seeks to determine the MCID/SCB/PASS values for the SDQ score in patients undergoing rotator cuff repair. This manuscript is well written and clinically relevant. While there are important limitations that need to be added in the discussion, this paper would be interesting and important for potential readers.

­          Introduction: I would recommend discussing a little more plainly how MCID/SCB/PASS are used in the literature, some readers may not be familiar with these. Specifically it may be beneficial to explain that these values are used in outcomes literature to asses the number of patients in a study who achieve a certain threshold to determine the efficacy of a particular treatment. Giving context for the importance of these values will help the readers understand why this study matters.

­          Lines 78-79: How many patients were initially enrolled who were lost to follow up?

­          Line 91 “An Italian study determined 91 the validity and reliability of SDQ score”: Which study was this, please cite

­          Lines 106-107: For the patients included in the “minimal improvement group” were these just those that said “a little better” or was it anyone who said “a little better or much better” as Those who answer “much better” also attained at least a Minimal improvement. Did you base your methods for determining MCID/PASS/SCB on previous literature, as there are numerous ways to calculate these values. I would recommend citing studies you based these methods on.

­          One limitation that is worth discussing is the generalizability of these values. The point of publishing something like this is so that other researchers can use the values you report to assess the success of a studied surgical intervention. As such it is important that the values reported in this study be generalizable so that other researchers can apply them to their patient population. With a limited number of patients included, the generalizability is questionable, which you briefly address. However some studies are finding that MCID/SCB/PASS may not be generalizable at all. For example (Wellington, Ian J., et al. "Substantial clinical benefit values demonstrate a high degree of variability when stratified by time and geographic region." JSES International (2022).) showed that there is no homogeneity between SCBs of 4 different commonly used shoulder outcomes scores when separating patients by time or geographic region for shoulder arthroplasty. I would recommend including the questionable generalizability of these values in the limitations/discussions.

Reviewer 2 Report

The present paper measures the MCID, SCB and PASS of the Shoulder Disability Questionnaire and provides relevant cut-offs for understanding the results of a surgical intervention for rotator cuff pathology. 

I would like to raise some issues regarding the selection of the patients and the methodology of the study.

There is luck of any information about the patients who were enrolled in the study. What was their pathology? What kind of surgery did they have?

Why did the authors choose the anchor question "Would you generally think that your health is at least good?" in order to determine the PASS scores? This is a generic question which is used to evaluate a disease specific response.

The choice of a scale from -2 to +2 for scoring responses could be confusing for the patient. The authors could use a more simple scale, ie. Likert scale.

Reviewer 3 Report

The object of this study is to identify accurate MCID, SCB, and PASS values in patients with shoulder disease for the SDQ score. In other words, authors would like to validate the SDQ score and to identify cut-off value that could be used in clinical research.

Overall, interesting paper that could enable other researchers to use those results in their everyday routine.

The main criticism is the relatively low sample size enrolled in the study. Moreover, a more exhaustive explanation of the “effect-size” value must be reported, I wonder if only a reference is clear enough for the readers (obviously for me is not enough). Authors, please explain more.

Do you assessed also the inter-rater variability, if not, this could be also a possible confounder, therefore discussed in the limitation section.

I appreciate a lot your statistical description, but I found it not so clear:

1)      The distribution methods used to find the thresholds of MCID were 0.5 SD (one-half of the standard deviation), the SEM (Standard Error Measure) and the MDC (Minimal Detectable Change). At least a reference was need, otherwise 0.5SD could be considered a “magic number”.

2)      I believe that something more accurate is need to explain the statistical approach, from line 122-129. This is the core of the study.

3)      Moreover, in the result you stated that, you do not have any Ceiling and floor effects, but in the statistical analysis I do not find anything that could explain how you assessed those effects.

Round 2

Reviewer 2 Report

Thank you for your kind replies. The authors addressed all the issues and therefore I agree to publish the paper in its present form.